# A Text Mining-Based Survey of Pre-Impressions of Medical Staff toward COVID-19 Vaccination in a Designated Medical Institution for Class II Infectious Diseases

**DOI:** 10.3390/vaccines9111282

**Published:** 2021-11-05

**Authors:** Yoshiro Mori, Nobuyuki Miyatake, Hiromi Suzuki, Setsuo Okada, Kiyotaka Tanimoto

**Affiliations:** 1Department of Hygiene, Faculty of Medicine, Kagawa University, Miki 761-0793, Japan; miyarin@med.kagawa-u.ac.jp (N.M.); tanzuki@med.kagawa-u.ac.jp (H.S.); 2Sakaide City Hospital, Sakaide 762-8550, Japan; hosp02@city.sakaide.lg.jp (S.O.); taka12ki05@gmail.com (K.T.)

**Keywords:** COVID-19, vaccination, text mining, impression

## Abstract

The present study investigated the pre-impressions of medical staff toward coronavirus disease 2019 (COVID-19) vaccination in a designated medical institution for class II infectious diseases in Sakaide, Japan using a text mining analysis. A total of 387 medical staff were surveyed on their pre-vaccination impressions toward the COVID-19 vaccine using an open-ended questionnaire from March 1st to 7th (the first survey) and from March 22nd to 28th (the second survey) at Sakaide City Hospital, Sakaide, Japan. A total of 296 people answered the question for the first time and 234 people answered for the second time among the 387 people. The vaccination rate was slightly lower for the younger generation than for the older generation. Before the first vaccination, the younger generation expressed concerns about side effects as well as a negative impact on pregnancy. However, before the second vaccination, there were fewer concerns regarding side effects and words of reassurance were also noted. Nurses expressed more opinions about side effects in both the first and second vaccinations than other medical staff. Concerns regarding side effects among medical staff decreased with the progression of COVID-19 vaccination. These data may provide useful information about the promotion of COVID-19 vaccination to the public, particularly in the young generation and women.

## 1. Introduction

Coronavirus disease 2019 (COVID-19), which originated in Wuhan city, Hubei Province, China, has spread worldwide in a short period of time since December 2019 [1,2,3]. In March 2020, the World Health Organization (WHO) declared a pandemic [4]. COVID-19 was detected in Japan from the beginning of 2020 [5], and infection control measures through behavioral changes, the so-called three Cs, to avoid (1) closed spaces with poor ventilation, (2) crowded places with many people nearby and (3) close-contact settings, such as close-range conversations, have been recommended [6,7].

Sakaide City Hospital [8], which is a designated medical institution for class II infectious diseases with a focus on infectious disease control, and other medical institutions in Japan have been taking a number of measures in accordance with government policies [9,10]. A vaccine, as a core strategy against COVID-19, was approved in Japan in February 2021 [11]; however, vaccination programs outside of Japan were initiated in 2020 [12,13]. At Sakaide City Hospital, the vaccination of medical staff began in early March 2021 [14]. However, in February 2021, limited domestic data were available on the COVID-19 vaccine.

Although medical staff were considered to have better access to information on the vaccine than the general public, they may still have had misconceptions and concerns.

A qualitative analysis using a text mining procedure provides useful information, similar to a conventional quantitative analysis. Yoshikawa et al. investigated childcare-related difficulties encountered by women who gave birth following kidney transplantation using the KH coder procedure [15]. Hasegawa et al. also clarified the potential factors associated with radiation-related anxieties caused by the Fukushima Daiichi Nuclear Power Plant accident [16]. Hemsley et al. reported that the purpose of sociotechnical research was to assess the impact of teaching individuals to use Twitter [17]. These findings demonstrated the usefulness of a qualitative analysis using KH coder.

There have been some studies on the attitudes of healthcare professionals and the general public toward COVID-19 vaccination [18,19,20,21]. However, there is no literature that has qualitatively evaluated by using text mining. We considered an increase in vaccination numbers to be possible by extracting and evaluating any issue at the start of the vaccination program.

Therefore, to obtain data to promote widespread vaccination in the future, we conducted a survey on the pre-impressions of medical staff at Sakaide City Hospital toward COVID-19 vaccination using a text mining analysis.

## 2. Materials and Methods

### 2.1. Subjects

Among the 387 medical staff at Sakaide City Hospital, Sakaide, Japan who agreed to participate in a survey of their pre-impressions toward each vaccination (COMIRNATY intramuscular injection, Pfizer Japan Inc., Tokyo, Japan), 296 (71 men and 225 women, 41.7 ± 12.1 years) (76.5%) from the first survey and 234 (64 men and 170 women, 41.6 ± 12 years) (60.5%) from the second survey were used in the analysis (Table 1).

### 2.2. Clinical Parameters

The sex, age and job title of enrolled subjects were obtained from self-reported questionnaire. Vaccination rate was also evaluated by age, sex and job title.

### 2.3. Questionnaire

Before each vaccination, an open-ended questionnaire was surveyed from March 1st to 7th (the first survey) and from March 22nd to 28th (the second survey) as follows: “What do you think of the COVID-19 vaccine?”. Questionnaires were distributed through intra-hospital e-mail, and responses were collected and analyzed. Sex, age and job title information were also included in the analysis.

### 2.4. Statistical Analysis

Data were expressed as the mean ± standard deviation (SD). The χ^2^ test was used to evaluate differences in vaccination rates among groups based on sex, age and job title, where *p* < 0.05 was considered to be significant. A quantitative analysis of impressions toward the vaccination was performed using text mining software (KH coder 3.0, Koichi Higuchi, Japan) and extracted words were converted into English. KH Coder produces a list of words ordered according to their frequencies and interrelationships as previously described as follows [22,23]: “Step 1: Extract words automatically from data and statistically analyze them to obtain a whole picture and explore the features of the data while avoiding the prejudices of the researcher. Step 2: Specify coding rules, such as “if there is a particular expression, we regard it as an appearance of the concept A”, and extract concepts from the data. Then, statistically analyze the concepts to deepen the analysis” [22]. We used a correspondence analysis, which is an analytical method that allows the relationship between words to be visualized in a scatter plot.

### 2.5. Ethics

Ethical approval for the present study was obtained from Sakaide City Hospital, Sakaide, Japan (2020-014, 15 February 2021).

## 3. Results

Table 1 shows the clinical characteristics of subjects enrolled in the present study. There were 296 subjects (39 medical doctors, 158 nurses, 38 other medical staff and 61 administrative staff) in the first survey and 234 (34 medical doctors, 121 nurses, 36 other medical staff and 43 administrative staff) in the second survey (Table 1).

The comparisons of vaccination rates stratified by sex, age and job title in the first and second surveys are shown in Table 2. Significant differences were observed in the vaccination rate with age in the first survey, with the rate of vaccination being lower in the younger generation (83.6%) than in the older generation. However, no significant differences were observed in the vaccination rate between men and women or among job titles in the first survey (Table 2). In the second survey, the vaccination rate significantly differed between men and women only.

Table 3 shows a list of frequently used words among nouns and adjectival nouns. In the first survey, the most common word among 5918 words was “side effects” followed by “vaccination”, “vaccine”, “worry” and “anxiety”. In the second survey, the most common word among 7614 words was “vaccination” followed by “anxiety”, “side effects”, “reaction” and “vaccine”. Therefore, the frequency of “side effects” decreased from the first to the second survey (Table 3). In addition, “reassurance”, which was not observed in the first survey, was noted as the 13th most frequent word in the second survey.

We performed a correspondence analysis to examine the relationships among words according to the manual in the previous reports [22,23] (Figure 1 and Figure 2). We evaluated the relationship between frequently used words and age (Figure 1). The words “side effects”, “scary”, “pregnancy”, “possible” and “future” were more frequently used among subjects in their 20s in the first survey. In the second survey, the words “pregnancy”, “possible” and “future” were not observed in this age group. The relationships between the frequently used words and job titles are shown in Figure 2. In the first survey, characteristic words were “early” and “reaction” in medical doctors, “side effects” and “vaccine” in nurses, “effect” in other medical staff and “concern” in administrative staff. In the second survey, characteristic words were “shoulder” in medical doctors, “scary” and “side effects” in nurses and “next” and “influenza” in other medical staff and administrative staff. Finally, we added the same text mining analysis with 209 participants who answered the questionnaire for the first and second time consecutively. We evaluated the relationship between the frequency used words and age (Figure 3) and obtained almost similar results.

## 4. Discussion

In the present study, we examined the pre-impressions of medical staff toward COVID-19 vaccination at a designated medical institution for class II infectious diseases using a text mining analysis. The vaccination rate was lower in the younger generation in the first survey and medical staff were less concerned about side effects with the progression of COVID-19 vaccination.

There are some studies about pre-impressions and attitudes toward COVID-19 vaccination. The majority of healthcare workers in Asia are willing to receive a COVID-19 vaccine, and the low potential risk of vaccine harm and pro-socialness are the main reasons [18]. Okubo et al. reported that the proportion of COVID-19 vaccine hesitancy among younger respondents was more than double that among older respondents. Female, living alone, low socioeconomic status and the presence of severe psychological distress were associated with hesitancy [19]. Yoda et al. showed that males showed less hesitancy toward being vaccinated [20]. Machida et al. showed that vaccine acceptance was lower in women, subjects aged 20–49 years and those with a low-income level [21]. In this study, the vaccination rate in younger generations at the first survey and women at the second survey was lower, as previously described [19,20,21]. In addition, we evaluated the pre-impressions toward COVID-19 vaccination by using text mining analysis and found the pre-impressions in specific groups, which was strengthened by the participants who answered the questionnaire for the first and second time consecutively.

Before COVID-19 vaccination, a fear of vaccines and anxiety about adverse reactions were inferred, similar to those previously reported for other vaccinations [24,25]. There were also concerns about pregnancy and the effects of vaccination on the future fetus, similar to influenza vaccination [26]. Okuhara et al. showed that it was more effective to target vaccine-hesitant individuals rather than outright vaccine refusers because their attitudes toward vaccination are more amenable to change [27]. Before the first vaccination, the infection control team, which plays a major role in infection control in hospitals, such as the treatment of infectious diseases and countermeasures against resistant bacteria [28], was not able to provide adequate information on the COVID-19 vaccine to medical staff. After the first vaccination, detailed information on the vaccine, such as its side effects, including effects on pregnancy, was provided. In addition, vaccination became available throughout Japan and information became more readily available, which appeared to be the reason for the change in impressions toward vaccination in the second survey. However, a significant difference was observed in the vaccination rate between the sexes in the second survey (men: 100.0% vs. women: 93.5%). Pulcini et al. and others showed that men were more likely to receive vaccines [29,30,31]. By examining the vaccination rates together with the text mining results, we thought it would be possible to develop an approach to increase vaccination rates in specific populations. More detailed information may need to have been provided to women before the second survey.

Strategies against COVID-19 have become a public health challenge and vaccination is considered to be key in Japan and worldwide [32]. Although the present results were obtained from medical staff, we consider the promotion of vaccination to depend on education and the provision of correct information to the public, particularly the young generation and women.

There are several limitations that need to be addressed. The present study was conducted at one local hospital. Therefore, the results obtained are not applicable throughout Japan. Furthermore, the subjects enrolled in the present study were medical staff, who are considered to be more health conscious than community-dwelling individuals. In addition, among 387 medical staff, the questionnaire was only completed by 296 subjects (76.5%) in the first survey and 234 (60.5%) in the second survey. Moreover, we were unable to evaluate impressions after vaccination. Nevertheless, the results obtained in the present study provide useful information that will promote COVID-19 vaccination in Japan.

## 5. Conclusions

By analyzing the questionnaire responses with KH Coder, we were able to extract the concerns of healthcare workers before vaccination. These data may be useful for promoting COVID-19 vaccination to the public, particularly the younger generation and women.

## Figures and Tables

**Figure 1 vaccines-09-01282-f001:**
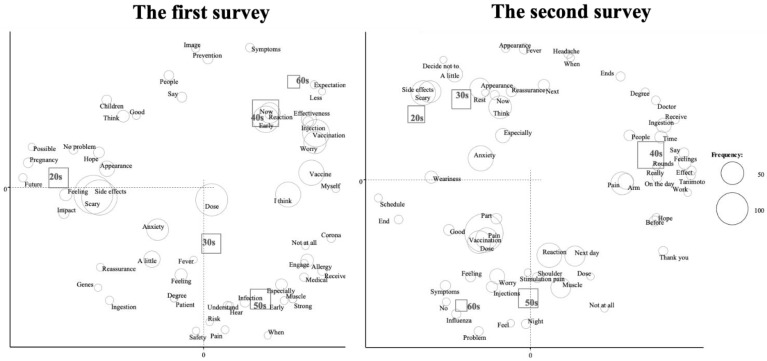
Correspondence analysis of the relationship between frequently used words and age.

**Figure 2 vaccines-09-01282-f002:**
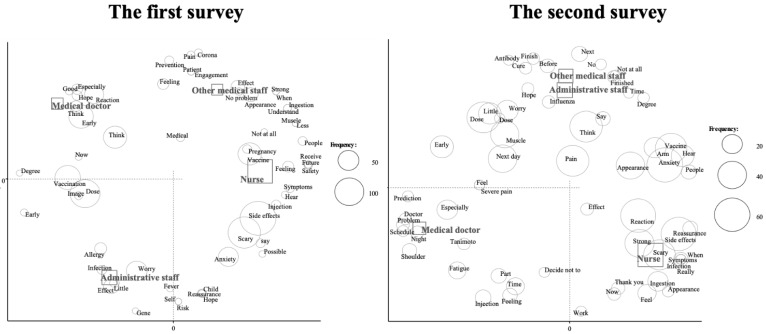
Correspondence analysis of the relationship between frequently used words and job titles.

**Figure 3 vaccines-09-01282-f003:**
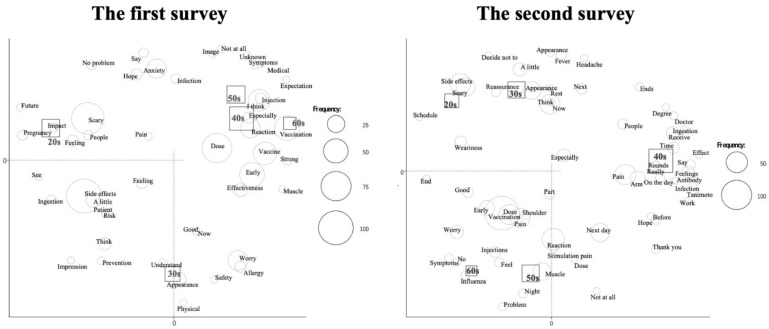
Correspondence analysis of the relationship between frequently used words and age in subjects who answered the questionnaire for the first and second time consecutively.

**Table 1 vaccines-09-01282-t001:** Clinical characteristics of enrolled subjects.

	Total		The First Survey		The Second Survey	
	Number of Subjects	%	Number of Subjects	%	Number of Subjects	%
**Age**						
20–29	91	23.5	67	22.6	50	21.4
30–39	83	21.4	61	20.6	49	20.9
40–49	120	31.0	94	31.8	77	32.9
50–59	61	15.8	50	16.9	38	16.2
60–	32	8.3	24	8.1	20	8.5
**Sex**						
Men	100	25.8	71	24	64	27.4
Women	287	74.2	225	76	170	72.6
**Job title**						
Medical doctor	51	13.2	39	13.2	34	14.5
Nurse	197	50.9	158	53.4	121	51.7
Other medical staff	48	12.4	38	12.9	36	15.4
Administrative staff	91	23.5	61	20.6	43	18.4

**Table 2 vaccines-09-01282-t002:** Comparison of vaccination rates (%).

	The First Survey		The Second Survey	
	Vaccination Rate (%)	*p*	Vaccination Rate (%)	*p*
**Age**				
20–29	83.6	**0.013**	90.0	0.176
30–39	91.8		93.9	
40–49	94.7		96.1	
50–59	98.0		100.0	
60–	100.0		100.0	
**Sex**				
Men	95.8	0.237	100.0	**0.037**
Women	91.6		93.5	
**Job title**				
Medical doctor	92.3	0.688	97.0	0.838
Nurse	91.8		94.2	
Other medical staff	97.4		97.2	
Administrative staff	91.8		95.3	

Bold: comparisons were performed using χ^2^ test.

**Table 3 vaccines-09-01282-t003:** List of frequently used words among nouns and adjectival nouns.

	The First Survey		The Second Survey	
	Word	Word Count	Word	Word Count
1st	Side Effects	148	Vaccination	147
2nd	Vaccination	74	Anxiety	63
3rd	Vaccine	63	Side effects	59
4th	Worry	49	Reaction	59
5th	Anxiety	46	Vaccine	49
6th	Reaction	41	Pain	47
7th	Effectiveness	22	Muscle	39
8th	Allergy	16	Worry	22
9th	Feeling	14	Ingestion	20
10th	Hope	11	Feeling	16

## Data Availability

Not applicable.

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
