# Peer review of "A Text Mining-Based Survey of Pre-Impressions of Medical Staff toward COVID-19 Vaccination in a Designated Medical Institution for Class II Infectious Diseases"

_vaccines, 2021, doi:10.3390/vaccines9111282_

Round 1

Reviewer 1 Report

This study investigates the pre-impressions of medical staff towards coronavirus disease 2019 (COVID-19) vaccination in a designated medical institution for class II infectious diseases in Sakaide, Japan using a text mining analysis. It can be accepted for the publication after some major revisions.

  1. Add the content of your questionnaire in the appendix.
  2. Figure 1 is not clear.
  3.  To help the reader, give more information about text mining analysis.
  4. Justify the choice of materials and methods used in this study.
  5. Give a comparison of your results with others existing in the literature.
  6. There are some typos. The authors should carefully read the manuscript

Reviewer 2 Report

In this study authors investigated the pre-impressions medical staff towards COVID 19 vaccination in a designated medical institution for class II infectious diseases in Sakaide (Japan) using a text mining analysis.

Althought several limitations such as the study was conducted at one local hospital and the subjects enrolled were medical staff who are considered to be more health conscious than community individuals conclusions are valid about the concerns of healthcare workirs before vaccination and the necessity of educating younger people and woman about appropriate vaccination.

Abstract

Well written and well structured.

Introduction

The introduction explain clearly the core strategy against COVID 19 in Japan.

Results

The results are supported by 3 tables and 2 figures which clarifies their compression of the manuscript.

References The bibliography is complete and up-to-date

Reviewer 3 Report

The manuscript titled “A Text Mining-based Survey of Pre-impressions of Medical Staff towards COVID-19 Vaccination in a Designated Medical Institution for Class II Infectious Diseases” proposed an investigation regarding pre-impressions of medical staff towards COVID-19 vaccination in a medical institution in Sakaide, Japan. A large sample of medical staff respondents was recruited in two different waves of vaccination and was administered an open-ended questionnaire. Text mining techniques were used to analyze data. Results showed that, before the first vaccination wave, younger participants expressed concerns about possible side effects and about a negative impact on pregnancy. Before the second vaccination wave, fewer concerns regarding side effects and some words of reassurance emerged. Nurses expressed more opinions about side effects in both waves of vaccination with respect to other medical staff participants. Authors discussed their results in light of previous literature highlighting strengths and limitations of their work.

I carefully read the manuscript, and I think it may be of interest for the readers of Vaccines. The topic is really relevant nowadays, and deserves more and more attention since the vaccination campaign is not over and we still need to convince a large part of population to take the COVID-19 vaccine. However, some major issues hamper the publication of the manuscript in its present form. Below you can find my comments and suggestions.

Abstract Section

Line 17: it is not clear in the abstract whether the participants are different or are the same people between the two waves of data collection. It seems like that the two sample sizes should sum up to 387, but 296+234 does not make 387. Please, clarify this point in the abstract.

Introduction Section

Lines 42-45: the rationale for conducting the present investigation is rather weak, there are neither reference to previous studies nor a clear motivation to conduct the present study. Please, reinforce the rationale by conducting proper bibliographic research on the topic of attitude about vaccination in medical staff and/or in the general population and describe it in a coherent theoretical framework.

Lines 46-47: scientific aims and specific hypotheses based on the rationale and on previous literature are lacking. What are your aims? Do you have specific hypotheses as well as expected results?

Materials and Methods

Lines 50-54: it is not clear why you chose to collect data in two waves of the same survey at the same participants. Please, clarify this point. Moreover, since 296+234 did not sum up to 387, this means the data regarding the two waves are not fully independent. I have the idea that the two waves of data collection correspond to the two doses of Pfizer vaccine, with a delay of 21 days between the each dose. If so, in the second survey you cannot talk of pre-impression, since a dose have already occurred and participants have had the time to observe side effects as well as other events. These events are no more a result of an attitude or a personal belief, but they are now real. If I am not wrong, please, how can you explain these argumentations?

Lines 64-66: It is the first time you talk about vaccination rates. What is the meaning of this analysis, why is it important for the study? To which aim is it related to?

Lines 70-71: Please report more information about data preparation (how many words there were at first, how many of them were deleted and why, et cetera) and about the parameters used for conducting the Correspondence Analysis.

Results Section

Lines 77-79: Those lines are not relevant, please remove them.

Lines 108-109: Table 3 should contains more words, since it refer to the main analysis of the manuscript. I think that the readers would like to see the occurrence of at least 10-15 words per survey.

Discussion Section

Lines 143-148: I think that the reasons the references for the usefulness of the qualitative analysis may be placed in the introduction section rather than in the discussion section.

I also think that the discussion section is lacking of a proper interpretation of the results. How can differences in pre-impressions between age groups and between job titles can be explained? Moreover, does make sense to compare the first and second survey since you have neither independent nor paired data? It would be better to retain only participants which answered to both surveys and compare them across time, knowing that they all received the first dose of vaccine.

Conclusions Section

Lines 189-190: the last sentence can be misinterpreted as “against minorities”. Also I don’t think this is the real message of the present study. A more complex framework could emerge if the methodology as well as the statistical analysis will be properly conducted.

Author Response

Please write down "Please see the attachment.

Round 2

Reviewer 1 Report

Now, the paper is well written and well organized.  Therefore, I recommend it for the publication in the journal. 

Reviewer 3 Report

I carefully read the resubmitted version of the manuscript titled “A Text Mining-based Survey of Pre-impressions of Medical Staff towards COVID-19 Vaccination in a Designated Medical Institution for Class II Infectious Diseases”. I think that the Authors did several changes according to referees’ comments, especially regarding introduction/discussion sections and results reporting, and that the present version has been improved with respect to the previous one.